# Tissue Oximetry Changes during Postoperative Dangling in Lower Extremity Free Flap Reconstruction: A Pilot Study

**DOI:** 10.3390/life13051158

**Published:** 2023-05-11

**Authors:** Anouk A. M. A. Lindelauf, Joep A. F. van Rooij, Loes Hartveld, René R. W. J. van der Hulst, Patrick W. Weerwind, Rutger M. Schols

**Affiliations:** 1Department of Cardiothoracic Surgery, Maastricht University Medical Center, 6202 AZ Maastricht, The Netherlands; 2Department of Plastic, Reconstructive and Hand Surgery, Maastricht University Medical Center, 6202 AZ Maastricht, The Netherlands

**Keywords:** lower extremity, free flap reconstruction, tissue oximetry, near-infrared spectroscopy, trauma, osteomyelitis

## Abstract

*Background:* Lower extremity free flap dangling protocols are still widely practiced, despite a paucity of evidence for their use. This pilot study investigates the use of tissue oximetry to provide further insight into the physiological effect of postoperative dangling in lower limb free flap transfer. *Methods:* Ten patients undergoing lower extremity free flap reconstruction were included in this study. Free flap tissue oxygen saturation (StO_2_) was continuously measured using non-invasive near-infrared spectroscopy. Measurements were performed on the free flap and contralateral limb during dangling from postoperative day (POD) 7 until 11, according to the local dangling protocol. *Results:* StO_2_ values measured in the free flap diminished to 70 ± 13.7% during dangling. This minimum StO_2_ was reached significantly later, and correspondingly the area under the curve (AUC) was significantly larger on POD 11 compared to the start of the dangling protocol on POD 7, reflecting an improving free flap microvascular reactivity. The dangling slope was equal between the free flap and contralateral leg. The reperfusion slope was significantly flatter on POD 7 compared to the other PODs (*p* < 0.001). Thereafter, no significant differences between PODs were observed. Patients with a history of smoking had significantly lower tissue oximetry values compared to non-smokers. *Conclusions:* The application of tissue oximetry during dangling provides further insight into the physiological effect (i.e., changes in microcirculatory function) of the free flap of the reconstructed lower extremity. This information could potentially be useful to either revise or disrupt the use of such dangling protocols.

## 1. Introduction

Extensive soft tissue defects of the lower extremity following trauma, oncological resection, infection or other wound healing problems can pose challenging problems, which often require a muscle or fasciocutaneous free flap for adequate coverage [1]. Free flap surgery involves a higher risk profile than other reconstructive procedures because the transplanted tissue relies on microvascular anastomoses for arterial inflow and venous drainage [2]. In addition, free flap transfers to the lower limbs are subjected to gravitational forces and associated with hemodynamic changes, which can cause edema and a decreased perfusion of the transplanted tissue, which could potentially compromise a successful reconstruction [3,4]. Therefore, after lower extremity free flap reconstruction, in most centers around the world patients follow a progressive dangling protocol to gradually habituate the flap to the effect of these gravitational forces [3,5,6]. Dangling protocols are still widely practiced, despite a paucity of evidence for their use [3,6,7].

Moreover, flap perfusion and viability are often determined by clinical observation, as well as results obtained by Doppler ultrasonography [6,8]. This is still the gold standard for assessing flap viability [5,9,10]. However, these measurements are only performed intermittently and rely on the experience of the assessor [11].

Tissue oximetry using non-invasive near-infrared spectroscopy (NIRS) has found applications for the continuous, real-time and objective assessment of flap perfusion in both experimental and clinical settings. Several previous studies suggest that tissue oximetry, during the first postoperative days, is capable of early detection of tissue hypoxia in different types of free flaps, resulting in a higher flap salvage rate than using only clinical observation [11,12,13,14,15].

Non-invasive tissue oximetry has also been used to investigate perfusion dynamics within the experimental field of lower limb free flap reconstruction during dangling [6]. A systematic review of flap survival following post-operative rehabilitation after lower limb free flap reconstruction elucidates changes in the perfusion parameters occurring throughout flap training [6]. However, the presented results are limited by the quality of evidence, and the direct physiological benefit of flap training may thus be inconclusive.

This pilot study investigates the use of near-infrared spectroscopy to provide further insights into physiological changes (i.e., tissue oxygenation changes) during postoperative dangling in lower extremity free flap reconstruction.

## 2. Materials and Methods

Institutional review board approval (METC 2019-1318) was obtained prior to conducting this prospective clinical pilot study. Written informed consent was provided before inclusion and the start of the measurements.

### 2.1. Study Subjects

Ten patients undergoing free flap reconstruction of the lower limb following either trauma or a salvage procedure in case of osteomyelitis at Maastricht University Medical Center (MUMC+) were included in this study. To identify patent vessels in the lower extremity, computed tomography angiography (CTA) was performed. Demographic patient data and the presence of risk factors such as hypertension, (history of) smoking, diabetes mellitus and/or peripheral arterial disease were collected.

### 2.2. Non-Invasive Tissue Oxygenation

Lower extremity free flap oxygen saturation (StO_2_) was continuously measured using near-infrared spectroscopy, based on the Lambert–Beer law. The proportion of light absorbed by oxyhemoglobin (HbO_2_) and deoxyhemoglobin (Hb) was determined using five wavelengths within the near-infrared spectrum (700–1100 nm) [16,17]. The ratio of HbO_2_ to the total concentration of Hb was used to quantify the oxygen content in the tissue [17,18].

In the current study, the FORE-SIGHT MC-2030 oximeter (Edwards Lifesciences Corporation, Irvine, CA, USA) was used to monitor non-invasive tissue oxygenation. A non-invasive disposable sensor was positioned on the lower limb free flap. A second sensor was placed at the same level on the contralateral, non-operated, leg. Sensors remained in position throughout the entire post-operative dangling protocol to allow maximal reproducibility. The sensor was immediately replaced in case of a loose sensor or other sensor-related problems. A flap dimension with a minimum of at least 50 by 30 mm was required to permit proper sensor placement, and to leave part of the skin-island of the free flap visible for clinical observation. For the first postoperative days (POD) the flap viability was assessed according to standard hospital care. For the first 24 h, clinical observation and Doppler measurements were performed every hour. On POD 2, this was conducted every two hours, on POD 3 every four hours and on POD 4 every eight hours. On POD 5 and 6, the free flap was examined only during the morning rounds. On POD 7, patients were permitted to start with flap training, i.e., dangling according to the protocol.

Tissue oximetry measurements were performed during dangling on postoperative day (POD) 7 until 11, according to the local dangling protocol:−POD 7: 2 times per day 10 min;−POD 8: 2 times per day 20 min;−POD 9: 2 times per day 30 min;−POD 10: 2 times per day 45 min;−POD 11: 2 times per day 60 min;−POD 12: unlimited dangling/discharge.

In addition, tissue oxygenation of the contralateral leg (control) was monitored as well. Measurements started 5 min before dangling to obtain baseline saturation values. After dangling, extremities were placed back in a horizontal position (reperfusion phase), whereafter measurements were continued until either the baseline value (minimal 5 min) or a steady state was reached.

All measurements were performed at the ward, by the same investigator and under standardized conditions. Moreover, results obtained by Doppler ultrasonography and the clinical observation of skin color and capillary refill were determined at the same time.

### 2.3. Statistical Analysis

Power calculations were not conducted, as this was a pilot study. Surgical details and physical characteristics were recorded in a comma separated values format, as well as postoperative complications related to flap viability that occurred in the postoperative period to hospital discharge. Absolute minimum StO_2_, ΔStO_2_, time to reach the minimum during dangling and time to recover after dangling were assessed. Next, the areas under the curve (AUC) during dangling and reperfusion were analyzed. The AUC was calculated for the first ten minutes of dangling and for the first five minutes during reperfusion, because these time periods were applicable in all measurements of the dangling scheme. Additionally, the dangling and reperfusion slopes were assessed as well.

Statistical analysis was performed to compare various subgroups. First, data were divided into two groups based on lower extremity free flap and contralateral leg. Subsequently, depending on data distribution (determined by the Shapiro–Wilk test) either an independent samples *t*-test or Mann–Whitney U-test was applied. For comparison between morning and afternoon measurements a paired samples *t*-test or Wilcoxon signed rank test was used.

Second, data of the free flap and contralateral leg were analyzed per individual day (POD 7 until POD 11). Depending on data distribution, a repeated measures ANOVA or Friedman test was applied. For comparisons between morning and afternoon measurements a paired samples *t*-test or Wilcoxon signed rank test was used.

Third, data were divided based on free flap type (anterolateral thigh [ALT] flap or free radial forearm flap [FRFF]). Depending on data distribution, either an independent samples *t*-test or Mann–Whitney U-test was performed.

Fourth, data were divided into two groups based on the anastomosis that was made (end to end vs. flow-through). For these analyses, either an independent samples *t*-test or the Mann–Whitney U-test was applied.

Finally, data were divided based on the presence of risk factors for potential flap loss. For these analyses, an independent samples *t*-test or Mann–Whitney U-test was performed.

For all tests, a two-tailed *p*-value of <0.05 was considered statistically significant. All analyses were performed using SPSS (version 28.0, SPSS Inc., Chicago, IL, USA) for Windows.

## 3. Results

Ten patients who underwent lower limb free flap reconstruction were included. The mean age at the time of surgery was 54 ± 18 years (Table 1). Patients had a mean body mass index of 26.1 ± 4.3 kg/m^2^. In most cases, osteomyelitis was the indication for surgery.

Two patients were diagnosed, before surgery, with hypertension, and one of those patients suffered from hypertension as well as diabetes mellitus. Two patients had peripheral arterial disease and three patients had a history of smoking (Table 2). All patients quit smoking at least one week prior to surgery.

Ten free flap reconstructions of the lower extremity were performed. In six procedures an ALT was used, and in four cases a FRFF transplantation was carried out. In eight patients, a CTA was made to identify patent vessels in the lower limb. In five patients, three vessels were intact; in two patients, two vessels were patent; in one patient, only a single vessel was available for the reconstruction. In almost all cases (90%), the posterior tibial artery was used for the anastomosis. The mean ischemic period of the free flap was 46 ± 17 min (Table 3).

The postoperative course of all free flaps was uneventful and there were no aberrant observations in Doppler signals and skin color (Figure 1). Capillary refill was normal in all cases before dangling, prolonged during dangling, and normal again after the reperfusion phase. Therefore, no alterations to the dangling protocol were necessary.

### 3.1. Free Flap vs. Contralateral Leg

As shown in Table 3, the StO_2_ measured in the free flap decreased to 70 ± 13.7% during dangling, whereas the StO_2_ measured in the contralateral leg decreased to a significantly lower value of 66 ± 8.4% (*p* = 0.016). Mean values for ΔStO_2_ from the free flap and the contralateral leg showed no significant difference: 19.9 ± 11.8% and 20.7 ± 8.0%, respectively (*p* = 0.597).

There were no significant differences in the AUC during dangling (free flap: 0.833 of optimal AUC, contralateral leg: 0.816 of optimal AUC; *p* = 0.237) and during reperfusion (free flap: 0.708 of optimal AUC, contralateral leg: 0.676 of optimal AUC; *p* = 0.983). The slope during dangling showed no significant differences (*p* = 0.151) between the free flap and the contralateral leg. During reperfusion, the slope of the contralateral leg was significantly steeper (*p* = 0.025) than the slope of the free flap.

InFigure 2, a tissue oximetry reading of an individual patient during dangling on POD 11 is shown. The time interval marked by number one indicates the baseline measurement, with both extremities in a horizontal, elevated position. Number two marks the dangling phase (both legs, in this case 60 min). Number three presents the reperfusion phase (both legs back in a horizontal, elevated position). At the X, the minimum StO_2_ was reached. The pattern in changes of StO_2_ observed in free flap was similar to the pattern of the contralateral leg throughout almost all measurements.

### 3.2. Individual Days

Data were subdivided per postoperative day (POD 7 until POD 11). The StO_2_ of the morning measurements of the free flap and contralateral leg were compared with the afternoon measurements. Only POD 7 showed a significant difference (*p* = 0.032) between the morning and afternoon measurement in the free flap (Table 4).

At the start of the dangling scheme (POD 7), there was a significant difference in time to reach the minimum StO_2_ between the free flap and the contralateral leg (*p* = 0.044). The reperfusion period showed no significant difference (Table 5). The morning measurements vs. afternoon measurements of ΔStO_2_ did also not show significance in any POD.

As shown in Table 6, the time to reach the minimum StO_2_ resulted in a significant increase between POD 7 and POD 11 in the free flap as well as the contralateral leg (*p* < 0.001, for both). However, POD 8 vs. POD 9 and POD 9 vs. POD 10 in the group of free flaps were not significant (*p* = 0.064 and *p* = 0.141, respectively). In the group of the contralateral leg, POD 9 vs. POD 10 (*p* = 0.388) and POD 10 vs. POD 11 (*p* = 1.000) were not significantly different. The reperfusion time after dangling was not statistically different in either the free flap or the contralateral leg between POD 7 and POD 11 (*p* = 0.055 and *p* = 1.00, respectively).

In Table 7, the *p*-values for the AUC and the slope of the free flap and contralateral leg during dangling are depicted.

As shown in Table 8, no significant differences in AUC during reperfusion were found. The slope during reperfusion on POD 7 was significantly different compared to the other PODs. No dissimilarities were noticed in the comparison of the other PODs.

When the data for reaching the minimum StO_2_ (Table 9) were divided into morning measurements and afternoon measurements, no significant changes were noticed either.

As shown in Table 10, the time of reperfusion after dangling between the morning measurement and afternoon measurement was significantly different on POD 7 in the free flap. No other significant changes were observed.

### 3.3. Free Flap Type

No significant differences in StO_2_, in AUC during dangling and reperfusion, and in dangling slope were found after comparison of the free flap type data, ALT vs. FRFF. The reperfusion slope of the ALT was significantly steeper than the slope of the FRFF (*p* < 0.001).

### 3.4. Anastomosis Type

No significant differences in StO_2_ in AUC during dangling and reperfusion or in the dangling/reperfusion slope were found after the comparison of types of anastomosis, end-to-end vs. flow-through.

### 3.5. Risk Factors

Three patients in the study population had a history of smoking. One patient quit smoking only one week prior to surgery. The StO_2_ of these patients during dangling were significantly lower in the flap (60 ± 16%) as well as in their contralateral leg (58 ± 13%) when compared to non-smoking patients (free flap: 73 ± 13%, contralateral leg: 68 ± 8; *p* = 0.001 for both). Of the AUC measurements, only the AUC during reperfusion was significantly different. In patients with a history of smoking, the AUC in the free flap as well as in their contralateral leg was smaller compared to non-smoking patients (*p* = 0.001 for both). The slope of StO_2_ values during dangling in the free flap for patients with a history of smoking compared to non-smoking patients was significantly steeper (*p* = 0.0043). The slope of the StO_2_ values during the reperfusion was significantly flatter in patients with a history of smoking (*p* < 0.001). These results were not seen in the contralateral leg. Hypertension, diabetes mellitus and/or peripheral arterial disease did not seem to cause any significant changes in StO_2_ (*p* > 0.05).

## 4. Discussion

In lower extremity free flap reconstruction, a variety of dangling protocols are used despite a paucity of evidence [3,6,7]. The amount of literature regarding postoperative care, and especially changes in the tissue oxygenation of lower extremity free flaps, is limited. Therefore, the aim of this pilot study was to investigate lower limb free flap tissue oximetry during postoperative dangling to provide further insight into the physiological effect (i.e., changes in tissue oxygenation) of such a protocol. We found that the minimum StO_2_ was reached significantly later and the area under the curve (AUC) was significantly larger on POD 11 compared to the start of the dangling protocol. No significant differences in reperfusion time and AUC between POD 7 and 11 were found. The dangling slope was equal between the free flap and contralateral leg. The reperfusion slope was significantly flatter in both the free flap and the contralateral leg when comparing POD 7 to the individual following PODs of the dangling protocol. Comparing the other PODs did not result in significant differences in the reperfusion slope. Patients with a history of smoking had significantly lower tissue oximetry values compared to non-smokers.

During postoperative dangling, StO_2_ values decreased to 70 ± 13.7%. This effect was most likely due to venous pooling, as the capillary refill during dangling was also prolonged. In the reperfusion phase (after re-elevation of the leg), the saturation and capillary refill were restored to their initial values. The lowest saturation was reached significantly later every POD in the free flap as well as in the contralateral leg, and the AUC during dangling became larger every POD. The pattern of changes in StO_2_ during dangling in the free flap were comparable to the changes in the contralateral leg. The measured slope of the curve (i.e., the first ten minutes of dangling) was not significantly different between the free flap and the contralateral leg. However, the StO_2_ values of the free flap were significantly higher than the StO_2_ values of the contralateral leg.

The results of this study are comparable to the results found in several previously published articles [4,19,20]. The desaturation in the free flap found in these studies was also comparable to our results. In the studies of Dornseifer et al. and Ridgway et al., the dangling time was set at five minutes per day (dangling period was POD 7 until discharge and POD 3 until POD 9, respectively). Both found a significantly higher minimum StO_2_ every following POD [19,20]. This is comparable with the study of Kolbenschlag et al. They started dangling at POD 6, three times per day for 5 min. At POD 9, the dangling scheme was completed at three times per day for 20 min. In the current study, the same mean minimum StO_2_ was found for every measurement during dangling, but this was reached at a later time point every POD. In our hospital protocol, dangling started with two times at 10 min on POD 7 until two times at 60 min on POD 11, which is considerably longer than in the previously published articles. These results suggest that using a dangling protocol after lower limb free flap surgery would indeed be beneficial to helping the free flapaccommodate to the gravitational forces and decreasing the occurrence of (partial) flap loss.

StO_2_ values started to decrease immediately after the start of dangling. This was not in line with the earlier-mentioned studies of Kolbenschlag et al. and Ridgway et al. [4,19]. They concluded that tissue oxygenation values initially increased after the start of dangling, assuming this would be the result of the increased inflow of oxygenated blood. Thereafter, venous pooling took over, resulting in a decrease of StO_2_ [4,19]. In the current study, this increase was not observed and was more in line with the study of Dornseifer et al. On POD 3 and 4, they observed this increase in StO_2_ as well, but on POD 5 this effect disappeared [20]. Furthermore, the saturation values of the contralateral leg were lower during the entire measurement than the saturation values of the free flap. This is in line with the study of Kolbenschlag [4]. They also observed a lower saturation in the contralateral leg. A possible explanation for the higher tissue oxygenation values of the free flap could be found in the formation of edema. NIRS values seem to be higher when swelling occurs [21]. To investigate whether this is the reason for increased StO_2_ values in this study, the tissue water index in the free flap and the contralateral leg needed to be investigated [22]. Another potential explanation for the discrepancy in StO_2_ values between the free flap and the contralateral leg could be vascular changes caused by endothelial damage as a result of the performed surgery [23].

The other discrepancies between our study and the available literature are probably the result of differences in the dangling protocols used and the diversity in the brands of devices. The values of different types of tissue oximeters cannot be compared, as concluded by Hyttel-Sorensen et al. [24].

To determine whether microcirculation is not affected during and/or after vascular procedures, a vascular (arterial) occlusion test (VOT) can be performed [25]. VOT monitors the hemodynamic response to induced ischemic stress [26]. Basically, three VOT-derived parameters can be identified: baseline (1), occlusion (2), and reperfusion (3), including the hyperemic area. The slope of the occlusion phase represents the rate of deoxygenation and reflects the microcirculation state, i.e., the rate of mitochondrial oxygen consumption and local tissue metabolism [27]. Thus, the lowest StO_2_ during the occlusion phase provides information about the absolute oxygen reserve in the regional microcirculatory system [26]. The reperfusion phase represents the vascular reactivity. With reduced vascular reactivity, i.e., microcirculatory dysfunction, the slope of the reperfusion curve will be flatter than in a healthy situation [28]. The last parameter is the hyperemic area, which reflects the tissue oxygen consumption [26]. Although the VOT is mostly performed by occluding the brachial artery, the physiological effects can be extrapolated to a provoked venous occlusion. During dangling, there is no genuine occlusion of the venous vessels but the occurrence of venous pooling can most likely induce ischemic stress (decrease in StO_2_). The pattern of StO_2_ values in this study is comparable with the changes found by Irwin et al. [29]. During dangling (ischemic stress), the StO_2_ drops and when the limbs are back in a horizontal position (reperfusion phase) the values gradually return to baseline. A clearly identifiable hyperemic area was absent during the current pilot study. A similar absence of the hyperemic area was noted by Irwin et al. [29]. Furthermore, the time of dangling in this study increased from 10 min on POD 7 to 60 min on POD 11. The lowest StO_2_ was reached later in every following POD, thus reflecting an improved free flap microvascular reactivity. Moreover, the reperfusion slope and time did not significantly change during the dangling scheme, implying an intact microcirculation after lower limb free flap surgery, in particular during the dangling protocol.

Congestion is one of the complications that can occur during dangling, and it can result in (partial) flap loss when it is not treated on time [30,31]. The forces that are responsible for the venous return of the blood during dangling are generally weaker than gravity, causing venous pooling in the free flap [30]. In the flow-through anastomosis (open circuit anastomosis), a high flow rate through the anastomotic site can be preserved even during dangling, in contrast to the end-to-end anastomosis (closed circuit). Miyamato et al. concluded that the influence of venous congestion on tissue oxygenation would be less using a flow-through anastomosis instead of end-to-end anastomosis [30,32]. However, their judgment of the extent of flap congestion was only based on clinical assignments in the first days post-operative, and not during dangling. In this study, no significant differences in tissue saturation values were found between free flaps with a flow-through anastomosis and an end-to-end anastomosis, suggesting that the superiority between both forms of anastomosis is scarce. Nevertheless, this study is not powered to distinguish between types of anastomosis, and additional research needs to be conducted to confirm the findings of this study.

In this study, ALT and FRFF flaps were measured. Almost no significant differences were found between both flaps. StO_2_ values before, during and after dangling were similar, as well as the AUC and slope during dangling. However, the slope during the reperfusion in the ALT was shown to be steeper than the slope of the FRFF. The ALT recovered significantly faster from dangling than the FRFF. In a study of Kolbenschalg et al., they also investigated different flap types; unfortunately not the FRFF. However, they also found the ALT to be the most adaptive free flap to the dangling scheme [4]. They hypothesized that, due to differences in blood supply between the different free flaps, the ALT would acclimate better to changes in perfusion and therefore be able to adapt better to the physiological effects of dangling.

Smoking is considered a risk factor for (partial) flap loss [33,34]. Nevertheless, in a recent study of Garip et al. no significant increase in risk of flap failure was found [35]. Although flap failure did not occur in this study, significant lower tissue oxygenation values were observed in patients with a history of smoking, and the slope during dangling (decrease in StO_2_ in the first 10 min of dangling) was also significantly steeper. The AUC during reperfusion was significantly lower and the slope during reperfusion was flatter in patients with a history of smoking, indicating a diminished perfusion of the tissue in the lower extremities during dangling due to microcirculatory dysfunction, which is in line with a study published in 2015 by Kolbenschlag et al. [36]. Tissue oxygenation will decrease because the carbon monoxide in cigarettes has a higher affinity with hemoglobin than oxygen [35]. In the earlier-mentioned study by Garip et al., and in two other recent publications, smoking showed a significant increase in wound dehiscence due to this diminished tissue oxygenation [35,37,38].

Peripheral arterial disease is closely related to smoking and is also a risk factor for (partial) flap loss [39]. In this study, two patients were included who suffered from peripheral arterial disease. All parameters of this study were not different in these patients compared to patients without peripheral arterial disease. The CTA from these patients showed three patent arteries for anastomosis. With a sufficient anastomosis, values are expected to be normal. In a study of Thai et al., they concluded that if the blood flow to the free flap is optimal, peripheral arterial disease would not negatively affect the success rate of the intervention [39].

In the current study, only ten patients undergoing free flap reconstruction of the lower limb were included. All cases were uncomplicated. Therefore, no conclusions regarding failed flaps could be made. Consequently, no cut-off values or ‘danger levels’ for StO_2_ values could be considered. However, the use of NIRS during dangling could have the potential to revise the protocol for postoperative care after lower limb free flap surgery. Based on the changes of tissue oxygenation values, changes in the dangling slope and changes in reperfusion slope, a more personalized dangling scheme for each patient could be provided. Or, if the conclusion of the ongoing multicenter study of Krijgh et al. [5] turns out to be that the use of postoperative dangling could be disrupted, implementing tissue oximetry using NIRS could then probably be beneficial to monitor the free flap and possibly prevent the free flap from overloading. A second limitation due to the multiple comparisons that were conducted, and the very small sample size, was the high possibility of a type I error. Furthermore, in this study only StO_2_ values were measured. In related studies, the regional hemoglobin, oxy- and deoxyhemoglobin, blood flow and tissue hemoglobin index (THI) in the free flap were also measured [3,20,36]. Unfortunately, these parameters are not incorporated in the Foresight device. Furthermore, investigating parameters such as THI and blood flow in combination with StO_2_ may increase the understanding of the physiological effect of dangling on the free flap.

Further research should consider the use of hyperspectral imaging (HSI), which has the potential to create an overview of tissue viability in free flaps [40,41].

## 5. Conclusions

We demonstrated that tissue oximetry with the use of NIRS during postoperative dangling provides further insights into the physiological effect (i.e., changes in microcirculatory function) on the free flap of the lower extremity. This information could potentially be useful to either revise or disrupt the use of postoperative dangling. Continuous monitoring of the free flap could have a positive effect on the duration of the dangling scheme, and therefore might reduce the hospitalization of patients after lower extremity free flap reconstruction in the future.

Nonetheless, a randomized control trial with a larger study population would be recommended to improve insights into the clinical use of tissue oxygenation measurements in free flaps during dangling.

## Figures and Tables

**Figure 1 life-13-01158-f001:**
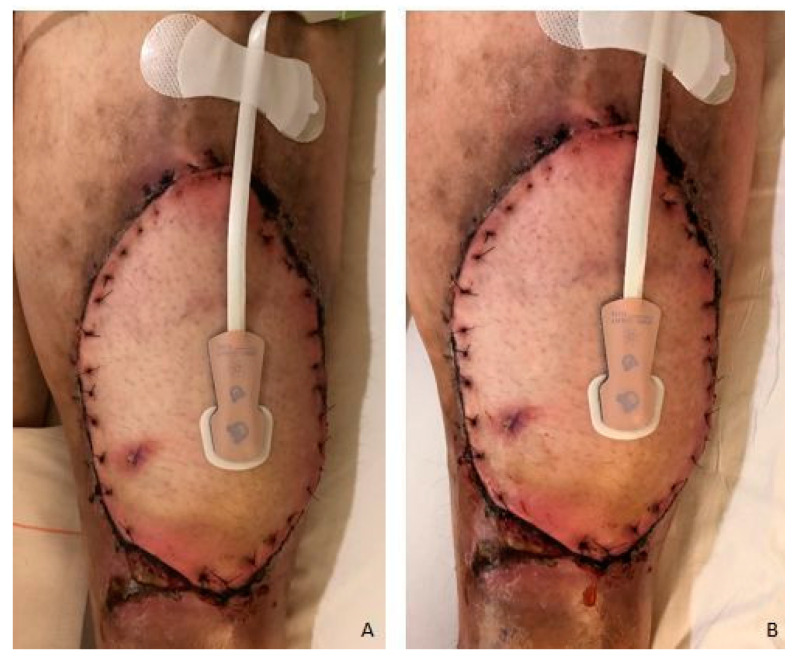
Free flap before (**A**) and after dangling (**B**).

**Figure 2 life-13-01158-f002:**
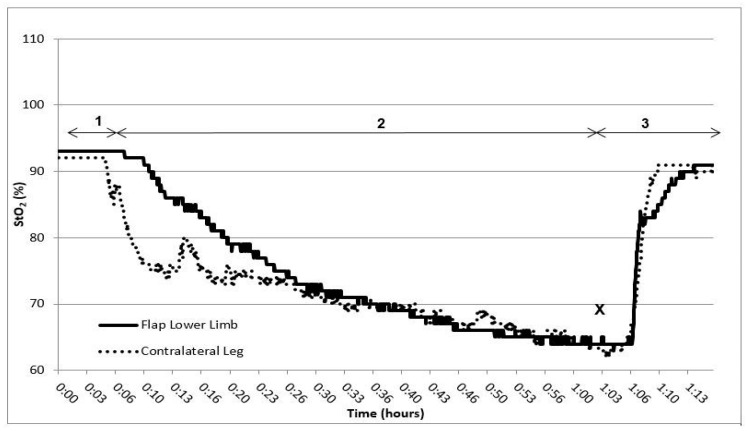
Tissue oximetry reading of an individual patient during dangling on POD 11. 1. Baseline measurement (extremities horizontal position), 2. Dangling phase (extremities vertical position), 3. Reperfusion (extremities horizontal position). X: Minimum saturation. StO_2_: tissue oxygen saturation.

**Table 1 life-13-01158-t001:** Patient characteristics.

Number of Patients	10
Male/female	7/3
Age (y)	54 ± 18
BMI (kg/m^2^)	26.1 ± 4.3
Osteomyelitis/trauma	7/3

BMI = body mass index. Values presented as absolute values or as mean ± SD.

**Table 2 life-13-01158-t002:** Presence of risk factors.

Risk Factor	Incidence
Hypertension	20% (*n* = 2)
History of smoking	30% (*n* = 3)
Diabetes mellitus	10% (*n* = 1)
Peripheral arterial disease	20% (*n* = 2)

**Table 3 life-13-01158-t003:** Free flap reconstruction characteristics.

Type of Reconstruction (*n* (%))		
	ALT	6 (60%)
	FRFF	4 (40%)
Vessel for anastomoses (*n* (%))		
	Posterior tibial artery	9 (90%)
	Dorsalis pedis artery	1 (10%)
Type of anastomosis (*n* (%))		
	End to end	6 (60%)
	Flow-through	4 (40%)
Ischemic time (min)		46 ± 17
Minimum StO_2_ during dangling free flap (%)		70 ± 13.7
Minimum StO_2_ during dangling contralateral leg (%)		66 ± 8.4

Time and StO_2_ in mean ± SD; ALT = anterolateral thigh flap; FRFF = free radial forearm flap; min = minutes; StO_2_: tissue oxygen saturation.

**Table 4 life-13-01158-t004:** StO_2_ during dangling in morning vs. afternoon.

POD	Free Flap			Contralateral Leg		
	Morning	Afternoon	*p*-Value	Morning	Afternoon	*p*-Value
7	77 ± 11	83 ± 13	**0.032**	69 [64–73]	70 [66–76]	0.386
8	72 ± 13	71 ± 13	0.849	68 [61–76]	67 [64–72]	0.932
9	67 ± 16	69 ± 11	0.613	66 [61–69]	64 [60–69]	0.813
10	65 ± 16	68 ± 12	0.571	67 [64–70]	67 [62–74]	0.623
11	65 ± 14	68 ± 13	0.201	64 [59–66]	63 [57–68]	0.878

POD = postoperative day. Data for free flap presented as mean ± SD. Data for contralateral leg presented as median (interquartile range [IQR]); bold *p*-values are statistically significant.

**Table 5 life-13-01158-t005:** Time to reach the minimum StO_2_ and time of reperfusion after dangling.

POD	Time to Minimum (Min)			Time of Reperfusion (Min)		
	Free Flap	Contralateral Leg	*p*-Value	Free Flap	Contralateral Leg	*p*-Value
7	6.0 ± 0.7	8.7 ± 0.8	**0.044**	2.6 ± 0.6	3.8 ± 0.7	0.198
8	15.9 ± 1.0	14.5 ± 1.0	0.957	4.3 ± 1.3	3.6 ± 0.6	0.621
9	23.0 ± 1.8	22.3 ± 1.8	0.407	3.7 ± 0.7	3.5 ± 0.6	0.928
10	28.4 ± 3.0	31.4 ± 3.1	0.479	5.3 ± 1.3	5.1 ± 1.3	0.934
11	46.0 ± 3.0	39.8 ± 4.5	0.452	5.7 ± 1.3	5.3 ± 0.8	0.856

POD = postoperative day; min = minutes. Data presented as mean ± SD; bold *p*-values are statistically significant.

**Table 6 life-13-01158-t006:** *p*-values for the time to reach the minimum StO_2_ and the time of reperfusion after dangling between PODs.

	Free Flap		Contralateral Leg	
	*p*-Value Time to Minimum (Min)	*p*-Value Time of Reperfusion (Min)	*p*-Value Time to Minimum (Min)	*p*-Value Time of Reperfusion (Min)
POD 7 vs. POD 8	**<0.001**	0.680	**0.004**	1.000
POD 7 vs. POD 9	**<0.001**	0.511	**<0.001**	1.000
POD 7 vs. POD 10	**<0.001**	0.512	**<0.001**	1.000
POD 7 vs. POD 11	**<0.001**	0.055	**<0.001**	1.000
POD 8 vs. POD 9	0.064	1.000	**0.035**	1.000
POD 8 vs. POD 10	**0.009**	1.000	**<0.001**	1.000
POD 8 vs. POD 11	**<0.001**	1.000	**<0.001**	1.000
POD 9 vs. POD 10	0.141	1.000	0.388	1.000
POD 9 vs. POD 11	**<0.001**	0.353	**0.011**	0.735
POD 10 vs. POD 11	**0.007**	1.000	1.000	1.000

Min = minutes; POD = postoperative day; bold *p*-values are statistically significant.

**Table 7 life-13-01158-t007:** *p*-values for AUC and slope during dangling between PODs.

	Free Flap		Contralateral Leg	
	*p*-Value AUC Dangling	*p*-Value Slope Dangling	*p*-Value AUC Dangling	*p*-Value Slope Dangling
POD 7 vs. POD 8	**<0.001**	0.287	**<0.001**	0.687
POD 7 vs. POD 9	**<0.001**	0.711	**<0.001**	0.911
POD 7 vs. POD 10	**<0.001**	1.000	**<0.001**	0.376
POD 7 vs. POD 11	**<0.001**	0.199	**<0.001**	0.370
POD 8 vs. POD 9	**0.004**	0.102	**<0.001**	0.879
POD 8 vs. POD 10	**0.011**	0.256	**<0.001**	0.811
POD 8 vs. POD 11	**0.005**	0.052	**<0.001**	0.573
POD 9 vs. POD 10	**0.007**	0.660	**<0.001**	0.601
POD 9 vs. POD 11	**0.003**	0.913	**<0.001**	0.794
POD 10 vs. POD 11	**<0.001**	0.569	**<0.001**	0.841

POD = postoperative day; bold *p*-values are statistically significant.

**Table 8 life-13-01158-t008:** *p*-values for AUC and slope during reperfusion between PODs.

	Free Flap		Contralateral Leg	
	*p*-Value AUC Reperfusion	*p*-Value Slope Reperfusion	*p*-Value AUC Reperfusion	*p*-Value Slope Reperfusion
POD 7 vs. POD 8	0.906	**0.019**	0.904	**0.024**
POD 7 vs. POD 9	0.845	**<0.001**	0.794	**0.030**
POD 7 vs. POD 10	0.287	**<0.001**	0.841	**0.009**
POD 7 vs. POD 11	0.948	**0.004**	0.911	**0.019**
POD 8 vs. POD 9	0.642	0.109	0.500	0.227
POD 8 vs. POD 10	0.278	0.278	0.616	0.163
POD 8 vs. POD 11	0.210	0.356	0.841	0.809
POD 9 vs. POD 10	0.113	0.868	0.546	0.286
POD 9 vs. POD 11	0.472	0.879	0.550	0.737
POD 10 vs. POD 11	0.586	0.795	0.295	0.157

POD = postoperative day; bold *p*-values are statistically significant.

**Table 9 life-13-01158-t009:** Difference in time to reach the minimum StO_2_ during dangling between morning and afternoon measurements.

POD	Free Flap			Contralateral Leg		
	Morning (Min)	Afternoon (Min)	*p*-Value	Morning (Min)	Afternoon (Min)	*p*-Value
7	6.2 ± 3.5	7.0 ± 3.5	0.231	8.4 ± 4.0	9.2 ± 2.5	0.603
8	16.8 ± 1.5	15.4 ± 6.1	0.753	14.5 ± 1.7	14.4 ± 6.1	0.924
9	21.8 ± 9.0	24.0 ± 6.0	0.521	23.3 ± 5.5	22.1 ± 9.0	0.750
10	27.0 ± 13.9	28.3 ± 11.4	0.319	26.0 ± 14.7	36.6 ± 8.5	0.149
11	42.3 ± 15.9	46.7 ± 13.1	0.430	43.9 ± 15.3	37.1 ± 21.4	0.354

Min = minutes; POD = postoperative day. Data presented as mean ± SD.

**Table 10 life-13-01158-t010:** Difference in time of reperfusion after dangling between morning and afternoon measurements.

POD	Free Flap			Contralateral Leg		
	Morning (Min)	Afternoon (Min)	*p*-Value	Morning (Min)	Afternoon (Min)	*p*-Value
7	3.7 [0.9–6.2]	0.9 [0.0–3.7]	**0.026**	3.2 [0.9–4.4]	3.6 [2.3–6.3]	0.214
8	2.9 [1.3–4.7]	2.4 [1.7–5.6]	0.506	3.7 [1.4–5.3]	2.8 [1.7–4.2]	0.648
9	3.7 [1.9–5.6]	3.1 [1.5–5.9]	0.853	4.0 [1.4–6.6]	2.9 [2.1–5.6]	0.327
10	4.7 [1.0–7.4]	4.1 [1.6–8.9]	0.167	2.2 [1.5–5.7]	3.5 [1.5–8.9]	0.799
11	5.3 [1.9–11.8]	2.7 [1.3–10.0]	0.168	5.9 [2.7–9.2]	4.3 [3.0–6.7]	0.262

Min = minutes; POD = postoperative day. Data presented as median (IQR); bold *p*-values are statistically significant.

## Data Availability

The data underlying the results presented in this paper are not publicly available at this time but may be obtained from the authors upon reasonable request.

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
