# Peer review of "Tissue Oximetry Changes during Postoperative Dangling in Lower Extremity Free Flap Reconstruction: A Pilot Study"

_life, 2023, doi:10.3390/life13051158_

Round 1
Reviewer 1 Report
The authors investigated tissue oximetry to provide further insight on the physiological effect of postoperative dangling in lower limb free flap transfer from POD 7 to 11. However, I can’t understand why the authors chose to perform tissue oximetry measurements and Doppler ultrasonography as well as clinical observation for assessing flap viability during gangling on POD 7 to 11 instead of from POD 1 to 7. In my experience, majority of the flaps have been survival on POD 7. I know the purpose of the author was to show the influences of dangling to the free flap reconstruction in lower limb. But it is no doubt that the influence of dangling to the flap is negative if the defects are not involved the joint. So in my opinion, this study did not offer the new information for the flap reconstruction. Also, the introduction was not consistent with the method and discussion. Furthermore, the numbers of the sample were too small to do statistical analysis.
Author Response
The authors are grateful for the remarks and are indebted to the reviewer for carefully reading their manuscript and providing detailed comments. A point-by-point feedback by the authors is given in the document added as appendix.

Reviewer 2 Report
The authors report a study on 10 patients undergoing free flap transfer to the LE and circulation measurement of their flaps while dangling.
The design is ok. However, there are several studies with more patients and better methodology (e.g. measurement with laser doppler in addition, positional challenges etc). Thus, it does not add that much in new insights.
Also, several issues arise.
Was wrapping performed? If yes in which manner? if no why?
10 patients in such a heterogenous collective ist just not enough power to distinguish between type of anastomosis etc.
how was the dangling procedure chosen?
discussion is well written and comprehensive.
conclusion:
"This information could be useful to either revise or disrupt the use of postoperative dangling."
which treshold would you recommend and based on which data?
Author Response
The authors are grateful for the remarks and are indebted to the reviewer for carefully reading their manuscript and providing detailed comments. A point-by-point feedback by the authors is given in the word document added as appendix.

Reviewer 3 Report
The authors conducted this study aiming to evaluate the use of NIRS in the field of postoperative tissue perfusion follow-up after free flap reconstruction when a dangling protocol was followed.
This study is very well designed. The introduction is very informative, providing the essential data needed. Moreover, the statistical analysis that followed was complete and correct, and the authors paid attention to the details (such as the number of the group, which defines how the normality of the data distribution is assessed using the Shapiro-Wilk or Kolmogorov-Smirnov test). In addition, the whole protocol followed was described in detail, which made the study reproducible. Regarding the results, there are some minor issues: 1) 3.1 paragraph: I could not understand the numbers referring to the AUC (e.g., lines 203-204: 69636%*min [41547-110628]). The AUC is expressed as a percentage because it reflects the area under the curve for the ideal point of maximum sensitivity and specificity (e.g., AUC = 88.3% or 0.883). Please define these results. 2) In tables 4–10, it is a little bit difficult to understand to which comparison each of the two p-values reported in each line refers. Please make this table easier to read by putting every p-value next to each comparison. 3) One disadvantage of the study that should also be reported is that there is a high possibility of type I error due to the multiple comparisons conducted and the very small sample.
Regarding the discussion There is nothing to mention. It is very extensive in describing the results of this study and comparing them with the international literature.
In conclusion, this is a nice job that adds more data to the limited ones found in the existing literature.
Author Response
The authors are grateful for the remarks and are indebted to the reviewer for carefully reading their manuscript and providing detailed comments. A point-by-point feedback by the authors is given and can be found in the uploaded file.

Round 2
Reviewer 1 Report
Thank you for your revision. Unfortunately, I still can’t understand the authors' opinion on the timing for monitoring flap during dangling on POD 7 to 11 instead of from POD 1 to 7. I am sorry to disagree with the authors. Also, I still think this manuscript should not be accepted.
Author Response
See previous. Editors only requested response on reviewer 3.
Reviewer 2 Report
I thank the authors for the changes made of which i fully approve.
Author Response
The authors are again grateful to the reviewer for carefully reading their manuscript, and of the approval to the changes that are made.